# Misinterpretations in Evaluating Interactions of Vanadium Complexes with Proteins and Other Biological Targets

**João Costa Pessoa *** and **Isabel Correia**

Centro de Química Estrutural and Departamento de Engenharia Química, Instituto Superior Técnico, Universidade de Lisboa, Av. Rovisco Pais, 1049-001 Lisboa, Portugal; icorreia@tecnico.ulisboa.pt
* Correspondence: joao.pessoa@tecnico.ulisboa.pt

**Abstract:** In aqueous media, $V^{IV}$- and $V^{V}$-ions and compounds undergo chemical changes such as hydrolysis, ligand exchange and redox reactions that depend on pH and concentration of the vanadium species, and on the nature of the several components present. In particular, the behaviour of vanadium compounds in biological fluids depends on their environment and on concentration of the many potential ligands present. However, when reporting the biological action of a particular complex, often the possibility of chemical changes occurring has been neglected, and the modifications of the complex added are not taken into account. In this work, we highlight that as soon as most vanadium(IV) and vanadium(V) compounds are dissolved in a biological media, they undergo several types of chemical transformations, and these changes are particularly extensive at the low concentrations normally used in biological experiments. We also emphasize that in case of a biochemical interaction or effect, to determine binding constants or the active species and/or propose mechanisms of action, it is essential to evaluate its speciation in the media where it is acting. This is because the vanadium complex no longer exists in its initial form.

**Keywords:** vanadium; proteins; DNA; fluorescence; binding constants; mechanism of action





## 1. Introduction

Many metal ions have a general tendency to interact with biomolecules, changing and/or modulating their properties and functions, therefore several of them are incorporated to perform crucial roles in organisms carrying out a wide variety of tasks [1,2]. Vanadium is a transition metal that is widely distributed on earth's crust, in soil, crude oil, water and air, so it is not surprising that it found roles in biological systems, being an essential element for many living beings.

Vanadium compounds may have oxidation states ranging from −III to +V, but $V^{III}$, $V^{IV}$ and $V^{V}$ are those of biological relevance. Vanadium ions bind to a broad range of biological compounds such as proteins, metabolites, membranes or other structures, and as with many other metal ions, a particular oxidation state may be stabilized by forming complexes with suitable ligands.

There are several enzymatic systems using vanadium in their active sites as relevant components for their function [3], and the complexes formed with several bio-ligands are important for the bio-distribution of vanadium [3–7]. For example, it is well established that vanadium binds to transferrin [8–15], this being relevant for its transport and bioavailability in blood. It is also well known that vanadium undergoes redox chemistry and other types of chemical transformations after administration [7,16–19], and understanding these processes is crucial to understanding the mechanisms of action.

In living systems V(IV) is normally in the form of $V^{IV}O$ [oxidovanadium(IV)], and V(V) as $V^{V}O$ [oxidovanadium(V)] or $V^{V}O_2$ [dioxidovanadium(V)], each of these moieties undergoing complex hydrolytic reactions which depend on pH, concentration and ionic strength [20–22]. Additionally, vanadium complexes, which we may designate as

[VO$_n$(L)$_m$], may also be involved in a wide range of reactions, interacting or forming complexes with metabolites, proteins and bio-ligands such as DNA, and these reactions depend on pH (typically in the range 6–8), and nature and concentrations of all species that may bind to vanadium. These include ligand L, H$_2$O, OH$^-$, H$_n$PO$_4$$^{-(3+n)}$ ions and any bio-ligand present. The actual speciation of vanadium in a system at a fixed pH depends on the total concentration of [VO$_n$(L)$_m$] present, on the type and concentrations of all species that may bind to vanadium and on the formation constants of the vanadium complexes formed.

If a vanadium compound, for example, V$^{IV}$OSO$_4$ or [V$^{IV}$O(acac)$_2$] (acac$^-$ = acetylace-tonato), is added to a particular biological system, for example, if it is placed in contact with cells, or administered to an animal, and if it exerts some biological effect, it is important to disclose which is/are the particular vanadium species that is responsible for the activity detected. Often the biological effect has been simply, and often wrongly, assigned to either V$^{IV}$O$^{2+}$ or [V$^{IV}$O(acac)$_2$], not taking into account the hydrolytic and/or other transformations that these species might have had once added to the biological system.

In other simpler approaches, binding constants of vanadium complexes to bio-ligands have been determined assuming that the complex maintains its integrity once added to the aqueous solution containing the bio-ligand, and often that is not the case. In fact, misinterpretations in evaluating interactions of vanadium complexes with proteins, bio-ligands and/or other biological targets are quite common in the scientific literature, and this text aims to emphasize that care must be taken when interpreting spectroscopic or other analytic information, particularly when examining data involving low concentrations of the vanadium complexes.

## 2. Discussion

### 2.1. Hydrolytic Behavior of Oxidovanadium(IV) Ions

Vanadium(IV) normally exists as V$^{IV}$O-species and is quite susceptible to hydrolysis and oxidation for pH > 3. The hydrolytic behavior of V$^{IV}$O$^{2+}$ ions in water has been studied [9,22–27], being well understood up to pH ca. 4. To quantify the several complex V$^{IV}$O-species that may form, we use the usual definition of formation constants:

$$\beta_{pqr}: \quad p\text{V}^{IV}\text{O}^{2+} + q\text{L} + r\text{H}^+ \leftrightarrows \left[\left(\text{V}^{IV}\text{O}\right)_p(\text{L})_q(\text{H})_r\right] \tag{1}$$

When addressing formation constants of hydrolytic species, the coefficient $q = 0$.

At low pH in aqueous solution oxidovanadium(IV) ions exist as [V$^{IV}$O(H$_2$O)$_5$]$^{2+}$. As the pH is increased, the V$^{IV}$O$^{2+}$ ions are progressively partly transformed into [V$^{IV}$O(OH)(H$_2$O)$_4$]$^+$ and [(V$^{IV}$O)$_2$(OH)$_2$(H$_2$O)$_n$]$^{2+}$, these normally represented as [V$^{IV}$O(OH)]$^+$ and [(V$^{IV}$O)$_2$(OH)$_2$]$^{2+}$. For pH > 3.5–4, assuming oxidation to V$^V$ is avoided, other oxidovanadium(IV) hydroxides form and, depending on the total vanadium concentration (C$_V$), V$^{IV}$O(OH)$_2$ may precipitate (solubility product ~10$^{-23}$) [9,23,24]. For pH >12 it is clear that the predominant species is [V$^{IV}$O(OH)$_3$]$^-$ (abbreviation of [V$^{IV}$O(OH)$_3$(H$_2$O)$_2$]$^-$) [25–27], but what happens in the pH range 4–12 is not well established, and depends on C$_V$ [9,22]. For pH > 5, the formation of a species with a V$^{IV}$O:OH$^-$ ratio of 2:5 was established [28], but clearly the vanadium species that form are oligomeric, [(V$^{IV}$O)$_2$(OH)$_5$]$_m$$^-$, the value of m depending on C$_V$. It was never clarified if oligomeric [(V$^{IV}$O)$_2$(OH)$_6$]$_n$$^{2-}$ forms or not at higher pH values, but as mentioned, for pH >12 the formation of monomeric [V$^{IV}$O(OH)$_3$]$^-$ was established [25–27]. Costa Pessoa, from several calculations based on visible and circular dichroism spectra of solutions contain-ing oxidovanadium(IV) and amino acids (L-Ala [29], L-Ser, L-Thr [30], L-Cys, D-Pen [31], and L-Asp [32]) for pH > 5 determined the values of the formation constants of [(V$^{IV}$O)$_2$(OH)$_5$]$^-$ and [V$^{IV}$O(OH)$_3$]$^-$ as log $\beta_{20\text{-}5} \approx -22.3 \pm 0.3$ and log $\beta_{10\text{-}3} \approx -18.2 \pm 0.2$, respectively.

Figure 1A,B depicts species distribution diagrams for the oxidovanadium(IV) system at two different C$_V$ values. It must be kept in mind that V$^{IV}$ forms the insoluble V$^{IV}$O(OH)$_2$ for pH > 4 if C$_V$ is higher than ca. 10$^{-4}$ M. At pH > 5 oligomeric vanadium species become relevant, and for pH > 7–8 the hydroxide dissolves, but this may take time, oxidation

to $V^V$ taking place if dioxygen is not carefully removed. It should be highlighted that for pH > 5–6 $V^{IV}O^{2+}$ ($[V^{IV}O(H_2O)_5]^{2+}$) does not exist as such, and that for $C_V < 10^{-5}$ M oxidovanadium(IV) ions are soluble, being mainly in the form of $[V^{IV}O(OH)_3]^-$ and $[(V^{IV}O)_2(OH)_5]^-$ (Figure 1).

At pH = 7, in $C_V$ conditions that may occur in biological media (from $10^{-7}$ to $10^{-4}$ M), the main species present are $[(V^{IV}O)_2(OH)_5]^-$ and $[V^{IV}O(OH)_3]^-$ (Figure 1C); therefore, in any type of calculations, e.g., determination of binding constants, the concentration of $V^{IV}O^{2+}$ cannot be taken as equal to $C_V$. Additionally, in determination of cytotoxicity, or other similar type of parameters, where an oxidovanadium(IV) salt is added to incubation media of cells, oxidovanadium(IV) ions will be partially or totally oxidized to $V^V$ ions, therefore the biological activity determined will not be due only to $V^{IV}$-species, but also to the $V^V$-species formed. The longer the incubation time, the more probable the participation of $V^V$-species. When testing $V^{IV}O$-salts using $C_V$ values higher than ca. 100 µM, if no precipitation of $V^{IV}O$-hydroxide is detected, this is because either a significant amount of $V^{IV}$ oxidized, or $V^{IV}$ is bound to ligands present in the incubation media, hence protected from oxidation. This should be highlighted when reporting the data to avoid misunderstandings about the identity of the active species.

## 2.2. Hydrolytic Behavior of Oxidovanadium(V) Ions

The hydrolytic behavior of oxidovanadium(V) ions in water has been studied mainly by using pH potentiometric and $^{51}V$ NMR spectroscopy measurements [20,21]. The system is complex and the equilibria (and values of formation constants) somewhat depend on the ionic strength and the salt employed to set it [20]. Here, we will consider the physiologically relevant (for blood serum) 0.150 M NaCl medium, and Figure 2 depicts species distribution diagrams of $V^V$ hydrolysis in several distinct conditions.

At the pH values relevant in common biological conditions, and at low concentrations, $V^V$ exists mainly as $H_2VO_4^-$ and $HVO_4^{2-}$, often referred as $VO_3^-$ or as mono-vanadate ($V_1$). At pH 7 and low $C_V$ values, (e.g., 10 µM, Figure 2B,C) $V^V$ divanadates ($V_2$) or tetra-vanadates ($V_4$) almost do not form, but they become important at higher V concentrations (e.g., 1 mM, Figure 2A,C). In cells, usual physiological vanadate concentrations are also too low to allow the formation of oligovanadates, but in the pH range 3–5.5 or in certain confined cell compartments, decavanadates ($[H_nV_{10}O_{28}]^{(6-n)-}$, $V_{10}$) may be relevant species [34–38]. If a ligand L, other than $OH^-$, is present in solution, $V^V$-L complexes may form, but the fraction of $V^V$ that is not bound to L will be involved in speciation similar to the exemplified in Figure 2.

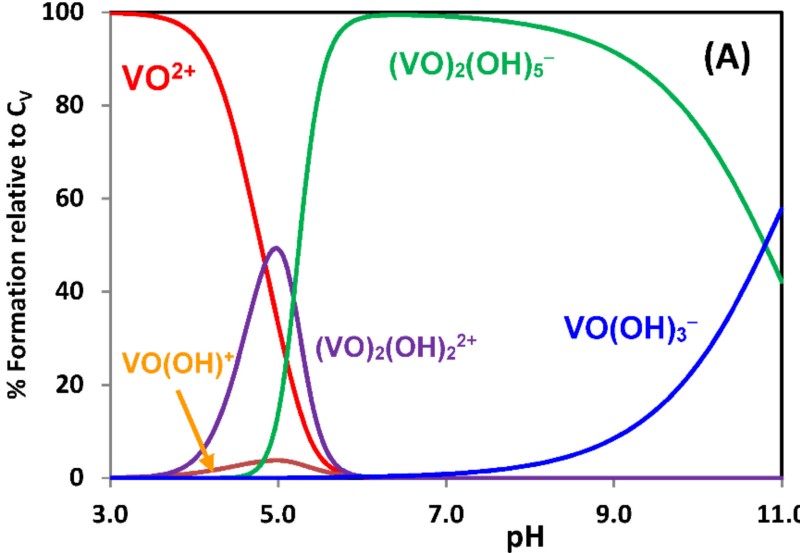

**Figure 1.** *Cont.*

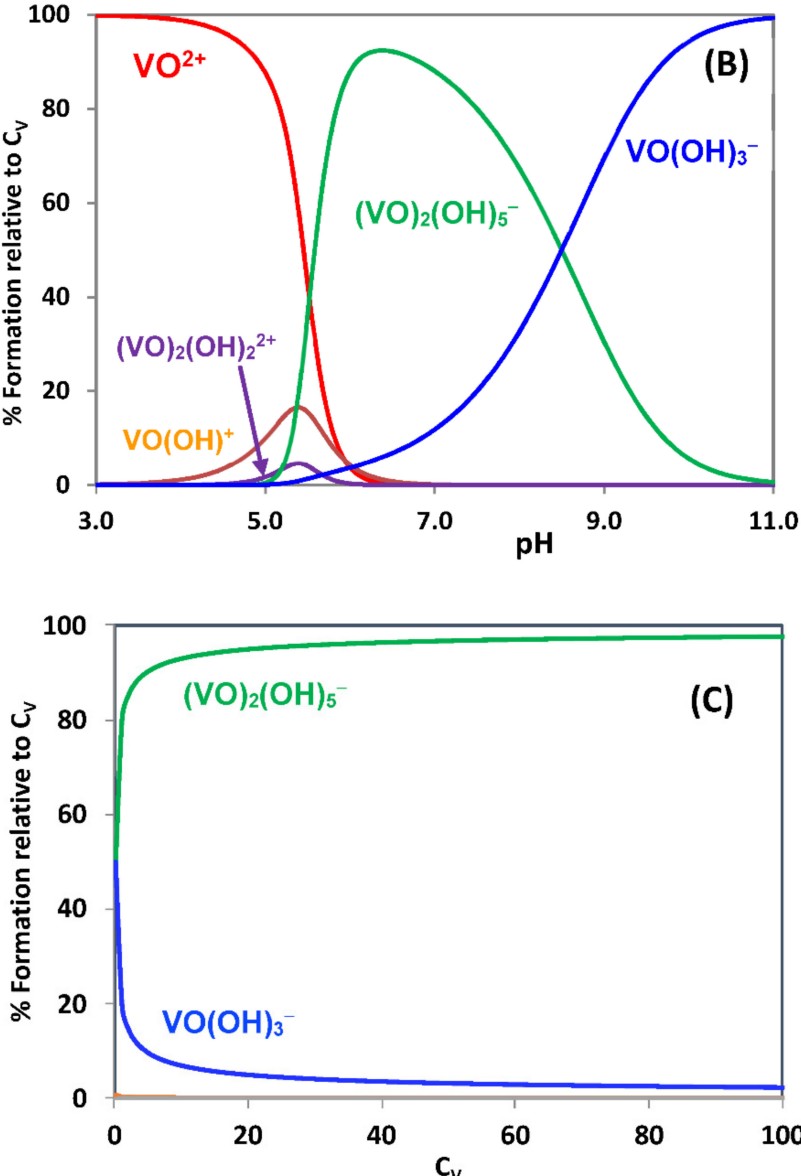

**Figure 1.** Species distribution diagrams, calculated with the computer program HySS [33], of $V^{IV}O^{2+}$ hydrolysis: (**A**) at $C_V = 2 \times 10^{-3}$ M, (**B**) at $C_V = 1 \times 10^{-5}$ M, in the pH range 3–11; (**C**) at pH = 7 with the total vanadium(IV) concentration ($C_V$) varying in the range 0.1 to 100 μM. The formation constants of the hydrolytic $V^{IV}O$-species were taken from [9,22]. In the diagram shown in (**A**), in the pH range 4–8 the product $[V^{IV}O^{2+}][OH^-]^2$ is higher than the solubility product of the hydroxide ($\sim 10^{-23}$), thus $V^{IV}O(OH)_2$ will precipitate.

### 2.3. Evaluation of Binding Constants of Metal Complexes with Bio-Macromolecules

To understand the biological activity of a metal complex or its transport in blood, it is important to evaluate how strong are its interactions with biological macromolecules such as proteins or DNA. Considering a [M(L)$_2$] complex such as $[V^{IV}O(acac)_2]$ or $[V^{IV}O(phen)_2]^{2+}$ (phen = 1,10-phenanthroline), this may correspond to the determination of the equilibrium constant of the following reaction at a particular pH value, e.g., pH = 7:

$$n\left[V^{IV}O(L)_2\right] + \text{biomolecule} \leftrightarrows \left[\left[V^{IV}O(L)_2\right]_n(\text{biomolecule})\right] \qquad (2)$$

$$K_2^{BC} = \frac{\left[\left[V^{IV}O(L)_2\right]_n (\text{biomolecule})\right]}{\left[V^{IV}O(L)_2\right]^n [\text{biomolecule}]}$$  (3)

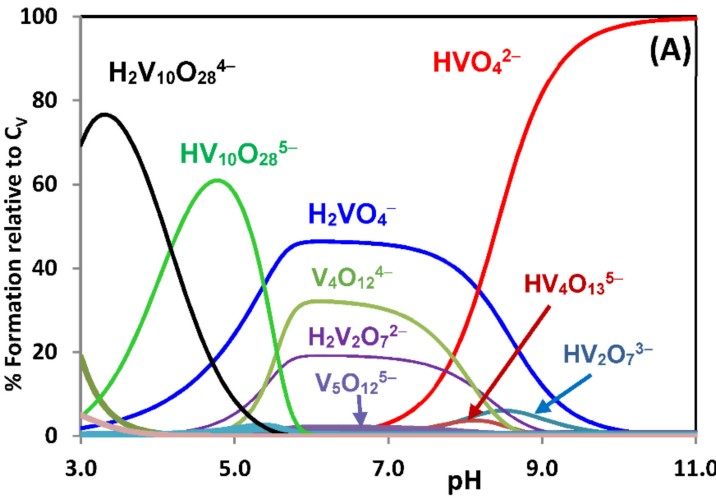

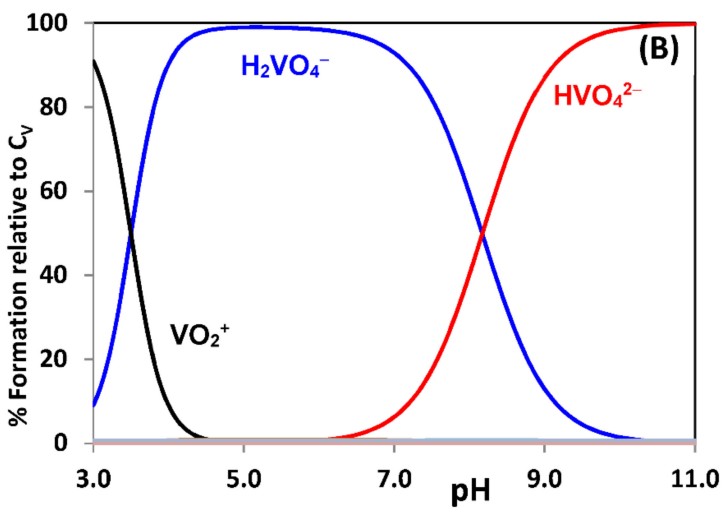

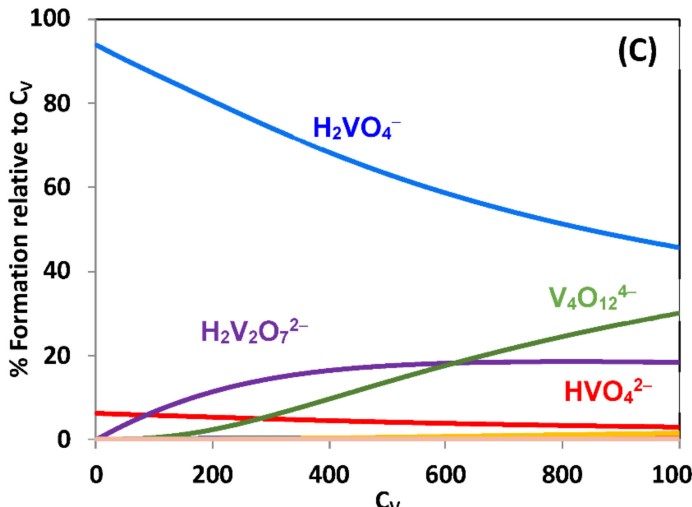

**Figure 2.** *Cont.*

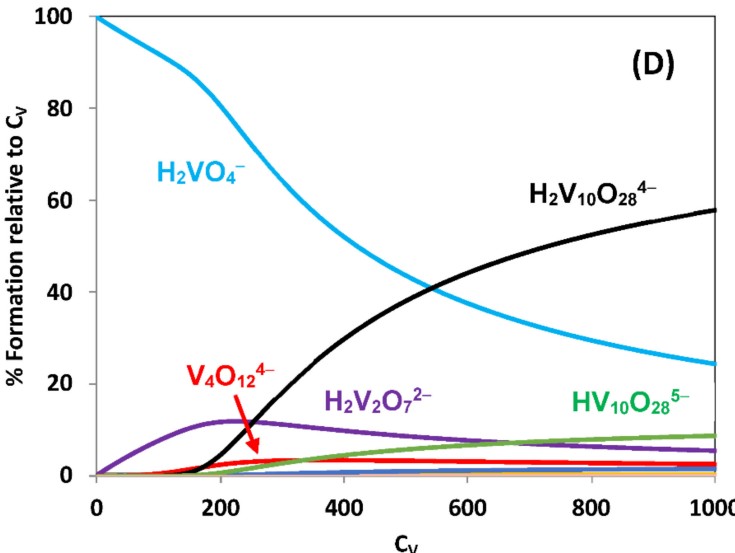

**Figure 2.** Species distribution diagrams, calculated with the computer program HySS [33], of $V^V$ hydrolysis: (**A**) at $C_V = 1 \times 10^{-3}$ M, (**B**) at $C_V = 1 \times 10^{-5}$ M, in the pH range 3–11; at pH = 7 (**C**) and pH = 5 (**D**) with the total vanadium(V) concentration varying in the range 1 to 1000 μM. The formation constants of the hydrolytic $V^V$-species were taken from [21].

Often this binding (or association) constant is designated by $K_a$ (or $K_b$) but these abbreviations should be avoided, as they are the symbols normally used to designate the dissociation constant of acids (or bases). We use $K_2^{BC}$ (and not $K^{BC}$) because we are assuming the binding of a [M(L)$_2$] complex (metal:ligand with 1:2 molar ratio); when considering a [M(L)] complex we will use $K_1^{BC}$ (metal:ligand with 1:1 molar ratio). The requirement of these distinct designations is clarified below. We also emphasize that the constants $K_1^{BC}$ and $K_2^{BC}$ correspond to the so called 'conditional' binding constants; their values depend not only on the pH of the solution but also on the type of buffer used. These may have components that have affinity for the metal ion, and thus may act as competitive ligands.

Most methods used to determine parameters of chemical interactions, the procedures to obtain binding constants may involve direct or indirect approaches. Using direct methods such as circular dichroism (CD) and/or UV–Vis electronic absorption spectrophotometry, complex-biomolecule binding constants may be determined accurately, but this typically involves the measurement of a great number of adequate data and the use of suitable computer programs. Indirect approaches such as fluorescence spectroscopy are normally less accurate but are much easier to use and require lower concentrations and less experimental data. In fact, fluorescence titrations must be performed at very low concentrations or preferably at concentrations in which the absorbance at the excitation wavelength is less than 0.05 (A < 0.05). Furthermore, the fluorescence response may no longer be linearly dependent on the light absorbed. Additionally, inner filter effects may also cause deviations from linearity at higher concentrations [39].

In the case of proteins, such as albumin, for which abundant studies are reported in the literature, the value of $K_2^{BC}$ has been determined mainly using fluorescence quenching measurements. The interaction of the metal complex and the protein may involve the quenching of the protein intrinsic fluorescence, due to Trp, Tyr or Phe residues; the interaction often gives rise to significant quenching of the protein's fluorescence and a high analytical signal. These methodologies involve several steps so that it might be confirmed that the fluorescence quenching is due to binding of a compound to the macromolecule and a static quenching process is operating, which is assumed to be due to the non-emitting complex [$V^{IV}O(L)_2$] associated to the macromolecule.

Titration experiments are usually carried out by fixing the concentration of one component, the biomolecule, while the concentration of the complex is varied. During the

course of the experiment, changes in the system are monitored, which are then plotted as a function of complex added. The resulting titration curve, known as a binding isotherm, is then fitted with a mathematical model derived from the expected equilibria to obtain the binding constant ($K^{BC}$). This model is usually developed from recognizing that the changes observed (e.g., quenching of fluorescence, $\Delta I$) are correlated to the concentration of the species $[([V^{IV}O(L)_2])_n(\text{biomolecule})]$.

When $n = 1$ in Equation (3), the total concentration of biomolecule and of the V-complex are the sum of the concentrations of free and associated forms, respectively:

$$[\text{biomolecule}]_T = [\text{biomolecule}]_{\text{free}} + \left[ \left[ V^{IV}O(L)_2 \right] \right) (\text{biomolecule})]] \tag{4}$$

$$\left[ V^{IV}O(L)_2 \right]_T = [([V^{IV}O(L)_2])(\text{biomolecule})]] + [V^{IV}O(L)_2]_{\text{free}} \tag{5}$$

It is not easy to measure the concentration of $[[V^{IV}O(L)_2])(\text{biomolecule})]$ (or $[V^{IV}O(L)_2]_{\text{free}}$ and $[\text{biomolecule}]_{\text{free}})$ but the knowledge of these is required to determine $K^{BC}$. However, these can be used to rewrite the concentration of associated form $[[V^{IV}O(L)_2])(\text{biomolecule})]$ (abbreviated as PC) as a function of the total concentrations, $[\text{biomolecule}]_T$ ($C_P$) and $[V^{IV}O(L)_2]_T$, ($C_C$) and of the binding constant, $K^{BC}$ [40].

$$[\text{PC}] = \frac{1}{2}\left( C_P + C_C + \frac{1}{K^{BC}} \right) - \frac{1}{2}\sqrt{\left( C_P + C_C + \frac{1}{K^{BC}} \right)^2 - 4\,C_P C_C} \tag{6}$$

The fluorescence intensity at a given wavelength ($I_\lambda$) of a given solution of bio-macromolecule and V-complex may be obtained from the molar fraction average of the fluorescence intensity of the individual species,

$$I_\lambda = \frac{[P]}{C_P} I_P + \frac{[PC]}{C_P} I_{PC} \tag{7}$$

where $I_P$ and $I_{PC}$ are the fluorescence intensities of the biomolecule in the absence of and presence of the V-complex, respectively. However, most reported $K^{BC}$ constants in the literature are obtained from linearizations, (linear regression methods) which are used due to its simplicity but with the power of modern computational methods should no longer be needed. There are at least two main problems connected with the use of these linearizations: (i) distortion of the experimental errors and (ii) assuming that $[C] \approx C_C$ (only valid when the biomolecule is in large excess) [38]. However, due to its prevalence in the literature and its simplicity we will use them as basis for the current discussion. Another problem with this methodology is that the binding may involve the formation of both a 1:1 and 1:2 ($V^{IV}O$:L) complexes, the treatment described below not being valid in these cases.

Conventionally, when molecules bind independently to a set of equivalent sites on a macromolecule, the equilibrium between free and bound molecules may be given by the following equation [41,42],

$$\log\left[(I_0 - I)/I\right] = \log K_2^{BC} + n \log [Q] \tag{8}$$

where $I_0$ and $I$ are the fluorescence intensities in the absence and presence of the quencher, respectively, $[Q]$ is the quencher concentration (in our case $Q = [V^{IV}O(L)_2]$), $K_2^{BC}$ is the binding constant defined according to Equations (2) and (3) and n is the number of binding sites per macromolecule. Using this methodology, if a linear relation between $\log((I_0 - I)/I)$ vs. $\log [Q]$ is obtained, values may be determined for $K_2^{BC}$ and n. As stated above, the value for $[Q]$ typically is taken as the total metal complex concentration (here $[V^{IV}O(L)_2]$), this being an approximation of the concentration of $[V^{IV}O(L)_2]_{\text{free}}$. This may be a wrong approximation, but is assumed in most publications. Moreover, as we will show, the fact

that a high analytical signal (e.g., a strong quenching of fluorescence) is measured, does not necessarily means a high accuracy for the calculated binding constant.

2.3.1. Binding to Proteins; Human Apo-Transferrin and $[V^{IV}O(acac)_2]$ as an Example

As an example we will consider the case of $[V^{IV}O(acac)_2]$ binding to apo-transferrin (apoHTF) [43]. Fluorescence quenching measurements were done at T = 298 K, pH = 7.4 and $\lambda_{ex}$ = 295 nm to study the binding of $[V^{IV}O(acac)_2]$ to apo-transferrin. A solution of apoHTF with concentration $1.02 \times 10^{-6}$ M was prepared, and a solution of $[V^{IV}O(acac)_2]$ was progressively added to get solutions with $C_V$ from 0 to $1.8 \times 10^{-5}$ M, i.e., with $[V^{IV}O(acac)_2]$:HTF ratios from 1 to ~18. Following the usual linearization methodologies (as described above) [41,42,44], it was assumed that the fluorescence quenching observed should be due to binding of the $[V^{IV}O(acac)_2]$ complex to apoHTF, thus a static quenching is operating which is due to the non-emitting $[V^{IV}O(acac)_2]$ bound to the protein. Using Equation (8), the values of $K_2^{BC} = 1.0 \times 10^4$ and $n = 1.15$ were obtained.

The apparent simplicity of this methodology led to its broad application in the scientific literature. However, fluorescence and its quenching involve an indirect measurement of the interaction of compounds with proteins such as apoHTF and serum albumins, as the Trp residues may not be close to the binding site, and binding at sites that do not significantly affect the fluorescence emission is also possible. Importantly, there are several requirements and possible pitfalls for the validity of use of fluorescence emission data to calculate binding constants, discussed in [41,44,45], emphasizing the most common errors made in their interpretation that should be considered.

Besides these requirements associated to the doubtful validity of the equations used, there are further aspects/issues related to the speciation of systems involving labile metal complexes at low concentrations which deserve further attention. In fact, not considering them has led to errors and misunderstandings. As shown below for the $[V^{IV}O(acac)_2]$–apoHTF system, although it is clear that there is binding of $V^{IV}O$-species to apoHTF that causes the fluorescence quenching, the methodology described, is not valid or reliable to calculate the binding constant of the equilibrium described by Equation (2).

In aqueous solutions containing Hacac and $V^{IV}O$ salts in 2:1 molar ratios and at mM concentrations, as the pH increases the following V-species predominate up to pH = 8: $[V^{IV}O(H_2O)_5]^{2+} \rightarrow [V^{IV}O(acac)]^+ \rightarrow [V^{IV}O(acac)_2]$. Taking the formation constants $\beta_{pqr}$ for the $V^{IV}O^{2+}$+acac system from [46], defined by Equation (1) with HL = Hacac, species distribution diagrams may be obtained for this system. Figure 3A depicts the speciation diagram calculated for total $V^{IV}O$ and acetylacetone concentrations of 3 and 6 mM, respectively. It is clear that in the pH range 6–8 $[V^{IV}O(acac)_2]$ is the main $V^{IV}$-complex present in solution. However, if the concentration of $[V^{IV}O(acac)_2]$ is lowered to ca. 2 μM, close to the concentrations used in the fluorescence quenching measurements in the system $[V^{IV}O(acac)_2]$ + apoHTF, the relative importance of most $V^{IV}$-species totally differs (see Figure 3B).

It is clear from Figure 3B that in the pH range 6–8, the relative concentrations of $[V^{IV}O(acac)]^+$ and $[V^{IV}O(acac)_2]$ in solution are low. At these low complex concentrations most of $V^{IV}$ is in the form of $[(V^{IV}O)_2(OH)_5]^-$ and $[V^{IV}O(OH)_3]^-$; therefore, in the conditions used to measure the fluorescence quenching, Equation (3) cannot be used with $[Q] = [V^{IV}O(acac)_2]$, and the fluorescence quenching observed may be due to several distinct species other than $[V^{IV}O(acac)_2]$.

To further understand how wrong it is to apply the methodology described above, we will obtain the speciation diagram for the system $V^{IV}O^{2+}$ + acetylacetone + apoHTF, assuming the formation of $[([V^{IV}O(acac)_2])_1(apoHTF)]$ with the value of $K_2^{BC}$ (= $1.0 \times 10^4$) obtained at pH = 7.4 based on data of fluorescence measurements. The value of log $\beta_{120}$ = 16.27 corresponds to the formation of $[V^{IV}O(acac)_2]$; therefore, the formation constant of $[([V^{IV}O(acac)_2])_1(apoHTF)]$, corresponding to the reaction of Equation (9) at pH = 7.4, is: $\beta_2^{BC} = K_2^{BC} \times \beta_{120} = 10^{20.27}$.

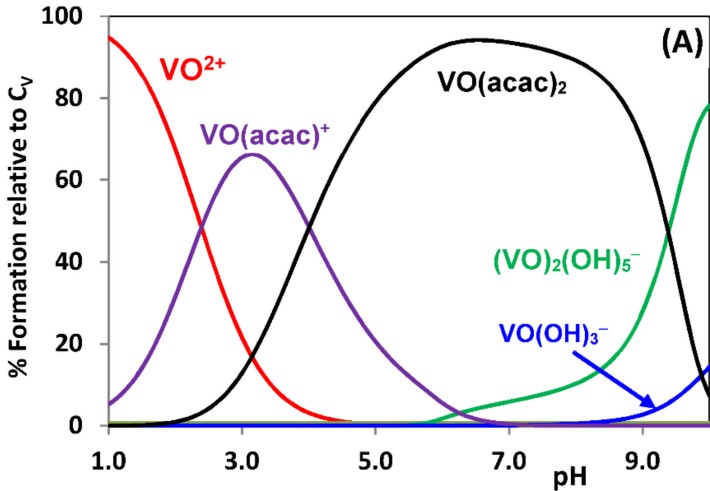

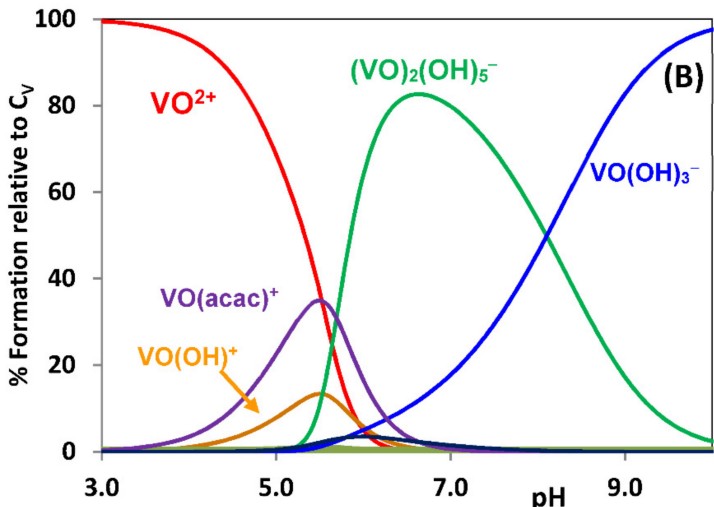

**Figure 3.** Species distribution diagrams, calculated with the computer program HySS [33], for total $V^{IV}O$ and acetylacetone concentrations of (**A**) 3 and 6 mM, respectively, and (**B**) 2 μM and 4 μM, respectively, considering the formation constants log $\beta_{110}$ = 8.73 and log $\beta_{120}$ = 16.27, reported in [46] and the $V^{IV}O$-hydrolytic constants: $[VO(OH)]^+$, $[(VO)_2(OH)_2]^{2+}$, $[(VO)_2(OH)_5]^-$ and $[VO(OH)_3]^-$ in refs. [9,22].

$$V^{IV}O^{2+} + 2acac^- + apoHTF \rightleftharpoons \left[\left(\left[V^{IV}O(acac)_2\right]\right)_1 (apoHTF)\right] \tag{9}$$

It is known that $V^{IV}O^{2+}$ binds to proteins, and the binding constants, at a particular pH value, may be defined as:

$$n\, V^{IV}O^{2+} + protein \rightleftharpoons \left(V^{IV}O\right)_n (protein) \tag{10}$$

$$\beta_n = \frac{\left[(V^{IV}O)_n(protein)\right]}{\left[V^{IV}O^{2+}\right]^n [protein]} \tag{11}$$

In the case of apo-transferrin $V^{IV}O^{2+}$ may form $[V^{IV}O(apoHTF)]$ and $[(V^{IV}O)_2(apoHTF)]$, and their formation constants were previously determined at pH 7.4 [15]: log $\beta_1$ = 13.4 and log $\beta_2$ = 25.2. The speciation may be calculated at pH = 7.4, see Figure 4, taking the con-

centration of apoHTF of $2 \times 10^{-6}$ M and varying the concentration of $[V^{IV}O(acac)_2]$ from $1 \times 10^{-6}$ M up to $20 \times 10^{-6}$ M, those used in the fluorescence quenching measurements.

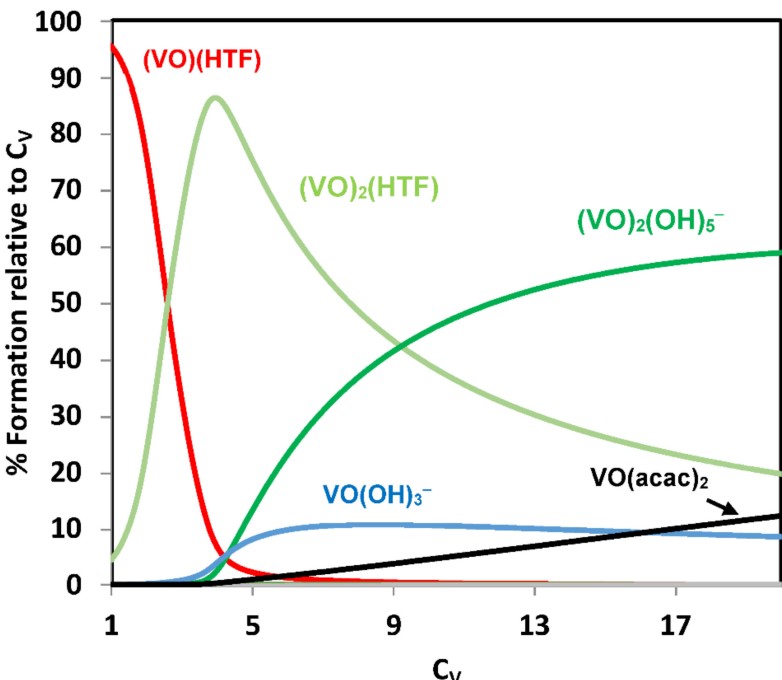

**Figure 4.** Species distribution diagram for the system $V^{IV}O^{2+}$ + acetylacetone + apoHTF, for [apoHTF]= $2 \times 10^{-6}$ M and $[V^{IV}O(acac)_2]$ in the range of 1 to 20 μM, at pH = 7.4, considering the formation constants reported in [9,15,22,46], and the formation constant of $([V^{IV}O(acac)_2])_1(apoHTF)]$ $(=10^{20.27})$ as obtained from fluorescence quenching methods. The calculations were carried out with the computer program HySS [33].

It is clear from Figure 4 that with a binding constant $K_2^{BC} = 10^4$ ($K_a$ in many publications) the amount of $[([V^{IV}O(acac)_2])_1(apoHTF)]$ formed is too low to have any relevant concentration at pH 7.4. Thus, the observed quenching of fluorescence cannot be solely due to the formation of $[([V^{IV}O(acac)_2])_1(apoHTF)]$, the value of $K_2^{BC}$ being clearly wrong.

Researchers should be aware that with values of association constants (log $K_1^{BC}$ or log $K_2^{BC}$) in the range of 4–8, typically found in the literature at pH 6–8, the relevance of binding of $[M(L)_n]$ (M = $V^{IV}O$) complexes to proteins such as apoHTF, BSA or HSA will be negligible in most cases. Similar conclusions may be extended to several other complexes of labile metal ions such as e.g., Cu(II), Zn(II) or Fe(III).

In the particular case of the $V^{IV}O^{2+}$ + acetylacetone + apoHTF system, only for values of log $K_2^{BC} > \sim 11$, which correspond to log$\beta_2^{BC} > 27$, the amount of $[([V^{IV}O(acac)_2])_1(apoHTF)]$ formed starts being visible in speciation diagrams at pH 7.4 (Figure 5). However, the system is much more complex as besides $[V^{IV}O(acac)_2(apoHTF)]$, other species probably form, e.g., $[V^{IV}O(acac)(apoHTF)]$, $[(V^{IV}O)_2(acac)(apoHTF)]$ and $[(V^{IV}O)_2(acac)_2(apoHTF)]$, and to properly address it and determine formation constants, spectroscopic techniques other than fluorescence should be used.

That was done for example in the $V^{IV}O^{2+}$ + maltol + apoHTF system at pH = 7.4, in which the formation constants were determined based on circular dichroism and EPR spectroscopy data, and calculated log β values of $[V^{IV}O(mal)(apoHTF)]$, $[(V^{IV}O)_2(mal)(apoHTF)]$ and $[(V^{IV}O)_2(mal)_2(apoHTF)]$ (mal = maltolato) were 17.7, 30.3 and 34.8, respectively. Similarly, in the $V^{IV}O^{2+}$+ dhp + apoHTF system (dhp = 1,2-dimethyl-3-hydroxy-4(1*H*)-pyridinone), the calculated log β values of $[V^{IV}O(dhp)(apoHTF)]$, $[(V^{IV}O)_2(dhp)(apoHTF)]$ and $[(V^{IV}O)_2(dhp)_2(apoHTF)]$ were 21.3, 33.0 and 40.3, respectively [15]. The systems $V^{IV}O^{2+}$ + picolinato + apoHTF, $V^{IV}O^{2+}$ + lactate + apoHTF, $V^{IV}O^{2+}$ + HSA and $V^{IV}O^{2+}$+

dhp + HSA and the corresponding formation constants were also determined based on EPR data [47].

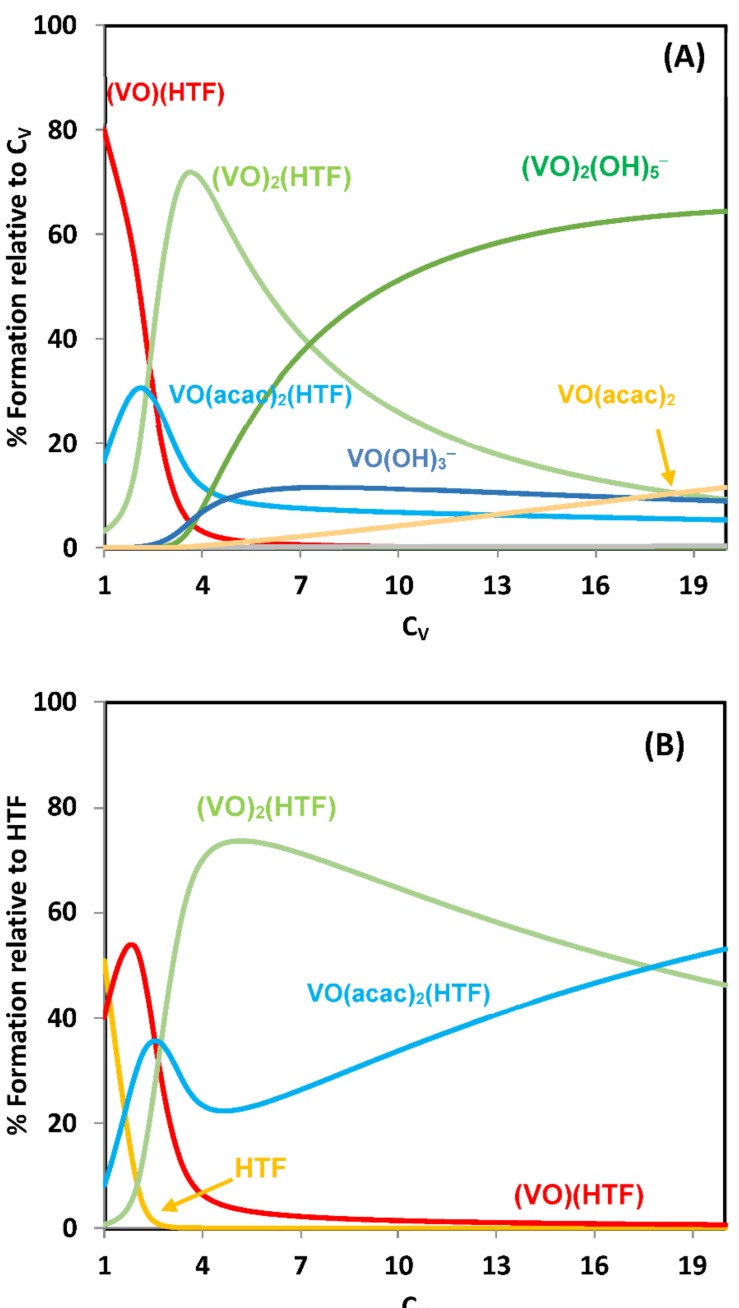

**Figure 5.** Species distribution diagrams for the system $V^{IV}O^{2+}$ + acetylacetone + apoHTF, for [apoHTF]= $2 \times 10^{-6}$ M and [$V^{IV}O(acac)_2$] in the range of 1 to 20 μM, at pH = 7.4, considering the formation constants reported in [9,15,22,46], and the formation constant of [($V^{IV}O(acac)_2)_1(apoHTF)$] (=$10^{27}$). (**A**) Representation of vanadium containing species; (**B**) Representation of apoHTF containing species. The calculations were carried out with the computer program HySS [33].

### 2.3.2. Binding to DNA

It is known that vanadate is possibly carcinogenic to humans. In vitro vanadate can induce DNA strand breaks in human fibroblasts [48], but it was also found that distinct cell lines may respond differently to NaVO$_3$, and that its carcinogenic role at low concentrations may result from stimulation of proliferation of tumorigenic cells [49]. $V^{IV}OSO_4$ was found genotoxic for normal and cancer cells being able to induce DNA damage in lymphocytes

(producing DNA single- and double-strand breaks) and in HeLa cancer cells (imposing only single strand breaks). Reactive oxygen species seem to be involved in the formation of DNA lesions [50].

Many studies reported cleavage/damage of DNA by either vanadium salts [48–52] or complexes [51,53–62]. It is known that several transition metals increase the oxidative stress in cells, which for vanadium has been mainly attributed to the generation of hydroxyl radicals by Fenton-like reactions promoted by physiological hydrogen peroxide [54,55]. Typically it is reported that vanadium(IV) reacts with $H_2O_2$ generating $HO^{\bullet}$ radicals, and these radicals may hydroxylate nucleobases and/or yield DNA strand breaks [51,52]. The involvement of singlet oxygen as the reactive species has also been suggested in some studies [56,63]. However, when reporting these reactions, the actual mechanisms of action of vanadium complexes is rarely accessed, most studies only stating if these processes are or not inhibited or activated by the presence of the ligand or of other chemical agents. In a study that included $V^{IV}OSO_4$, as well as various V-complexes (with picolinic acid and maltolate) and different oxidants, only $V^{IV}OSO_4$ showed ability to modify a double stranded 167-bp DNA fragment in the presence of $KHSO_5$. The mechanism proposed by the authors involved the oxidation of $V^{IV}$ to $V^V$ and the formation of a "caged" sulfate radical in the presence of $KHSO_5$. DNA cleavage at the guanine base was also proposed [64].

Notwithstanding, when reporting the action of metal complexes it is important to fully understand the role of the metal, of the ligand and of other species present. For example, the nuclease activity of $[V^{IV}O(acac)_2]$ and of several derivatives was studied by several techniques [54,55]; the mechanism was shown to be oxidative and the DNA cleavage mainly due to the formation of reactive oxygen species (ROS). Hydrolytic cleavage of the phosphodiester bond was also observed, but at much slower rate and did not compete with the oxidative route. Noteworthy, it was shown that the generation of ROS was much higher in experiments where phosphate was used as buffer, this being attributed to the formation of a mixed-ligand/mixed-valence complex containing phosphate, $[(V^{IV}O)(V^VO)(acac)_2(H_nPO_4{}^{n-3})]$ [54]. These studies focused on $[V^{IV}O(acac)_2]$ but in fact could/should be applied to other vanadium compounds, emphasizing the requirement of carefully accessing the speciation of the systems, which in this particular case is modified by the presence of phosphate. As phosphate is present at variable concentrations in most biological systems, this may have important implications in the interpretation of the biological activity of vanadium compounds. The extent of this effect depends on the phosphate:vanadium ratio and the formation of mixed-ligand species such as VO-L-phosphate (L = acac), which strongly enhances the oxidative stress, allegedly caused by $[V^{IV}O(acac)_2]$, most probably is not restricted to this particular vanadium system.

Bernier et al. investigated the nuclease activity of V(IV)-complexes of hydroxysalen derivatives under oxidative or reducing conditions. In the absence of activating agents, none of the complexes induced DNA cleavage; while in the presence of mercaptopropionic acid (MPA) or oxone all induced DNA modifications. The complexes reacted with DNA at guanine residues in the presence of oxone, and the mechanism proposed, based on spin-trapping EPR experiments, involved the oxidation of $V^{IV}$ to $V^V$, the production of $SO_4{}^{-\bullet}$ and $SO_5{}^{-\bullet}$ radicals via a redox reaction and the trapping of the sulfate radicals by the metal through the formation of a "caged" radical [58].

Besides the ability to cleave/damage DNA, the binding interactions of vanadium complexes to DNA have also been studied by different techniques; for example electronic absorption [56,59–62], circular dichroism [60–62], and fluorescence emission have been used frequently [42,60–62,65]; other techniques such as atomic force microscopy [66–69], viscosity [56,59–61,69] and have also been applied. $^{51}V$ NMR, [53,54] capillary electrophoresis and Fourier transform infrared difference spectroscopy [70] has also been used, but much less frequently, as well as DNA melting, applied to evaluate e.g., binding and/or DNA crosslink formation by the complexes [56,59,60,71].

These studies are normally done at pH ~7 and, as emphasized above, in most cases extensive hydrolysis of the complexes is expected. While many studies have confirmed the

binding of $V^{IV}O^{2+}$ and $V^V O_2^+$ to proteins such as HSA, BSA, apoHTF and immunoglobulin G (IgG), and formation constants determined [12,15,72–75], not much is known about the binding of these ions to DNA.

To the best of our knowledge, the only literature study on the binding $V^{IV}O^{2+}$ and $V^V O_3^-$ ions to *calf thymus* DNA in aqueous solutions at physiological pH, used capillary electrophoresis and FTIR spectroscopy. The authors reported that $V^{IV}O^{2+}$ binds DNA through guanine and adenine N-7 atoms and the phosphate groups with apparent binding constants of $8.8 \times 10^5$ M$^{-1}$ and $3.4 \times 10^5$ M$^{-1}$ for guanine and adenine respectively. The $V^V O_3^-$ ion shows weaker binding through thymine, adenine, and guanine bases, with $K^{BC} = 1.9 \times 10^4$ M$^{-1}$ and no interaction with the backbone phosphate moieties was found. The authors also reported a partial B-to-A DNA transition upon $V^{IV}O$–DNA binding. Again, the speciation occurring at physiological pH for V(IV) and V(V) species was not taken into account [70]. For DNA this is particularly relevant since it contains negatively charged phosphate groups that may participate in electrostatic binding to vanadium. Upon dissolution of $V^{IV}O^{2+}$ at physiological pH negatively charged species are formed— $[(V^{IV}O)_2(OH)_5]^-$ and $[V^{IV}O(OH)_3]^-$-and these have to be taken into account.

However, even if we assume that vanadium(IV or V) ions do not bind to DNA, when evaluating the binding of $[V^{IV/V}O_n(L)_m]$ complexes to DNA, to assume that the complexes do not hydrolyze at low concentration in aqueous solvents, is an oversimplification that normally is not acceptable.

### 2.4. Behavior of Metal Complexes when Added to Incubation Media of Cells

Recent studies regarding applications of vanadium compounds in therapeutics have been mainly focused on their anti-cancer and anti-parasitic potential and many vanadium complexes have been tested, mainly by in vitro studies [8,76]. Within these fields several polypyridyl complexes of V(IV), as well as mixed-ligand complexes containing polypyridyl ligands have been prepared and their anti-proliferative activity, cytotoxicity and ability to induce apoptosis tested in vitro, normally displaying high activities [8,77–96]. Possible biological targets and mechanisms of action of V-phen compounds have been discussed, and interaction with DNA has been typically considered as relevant [8,79].

$V^{IV}O$-complexes with polypyridyl ligands also hydrolyse and/or change their composition at low μM concentration when compared with mM concentrations [95,96], and this is clearly demonstrated in Figure 6 for the $V^{IV}O^{2+}$ + phen system. Decomposition of the $V^{IV}O$-complexes with polypyridyl ligands leads to the release of the free ligands, which by themselves are biologically active and cytotoxic [95–103].

Metal complexes may react with cell culture media components. This is important for their in vitro biological activity, [104] particularly for labile metal ions, but has been normally neglected in studies reporting cytotoxicity of vanadium compounds, as well as of other metal complexes. The role of the ligands in the cytotoxicity of $[V^{IV}O(OSO_3)(Me_2phen)_2]$ (Me$_2$phen = 4,7-dimethyl-1,10-phenanthroline), and of a few related $V^{IV}O$-complexes, was investigated by the group of Lay and co-workers [95]. These researchers reported cytotoxicity assays with human lung cancer A549 cells and tested the stability of the complexes in aqueous solutions as well as in cell culture media. They concluded that the high cytotoxicity at 72 h incubation of the free ligands and corresponding $V^{IV}O$-complexes is due to the free ligands upon decomposition of the complexes in cell culture medium.

More recently, Nunes et al. [96] reported cytotoxicity studies of a group of $V^{IV}O$-polypyridyl complexes at several incubation times against three different types of cancer cells. The compounds addressed were $[V^{IV}O(OSO_3)(phen)_2]$ $[V^{IV}O(OSO_3)(Me_2phen)_2]$, and $[V^{IV}O(OSO_3)(amphen)_2]$ (amphen = 5-amino-1,10-phenanthroline). We will globally designate these 1,10-phenanthroline compounds by Xphen. Upon incubation for 72 h with several types of cells, the cytotoxicity of all compounds was approximately equal, while at incubation times of 3 and 24 h, where the IC$_{50}$ values measured are much higher, the complexes show significantly higher activity than the free ligands. This difference in

behavior is probably due to the distinct type of speciation taking place in cell media. Next, we shortly discuss these observations.

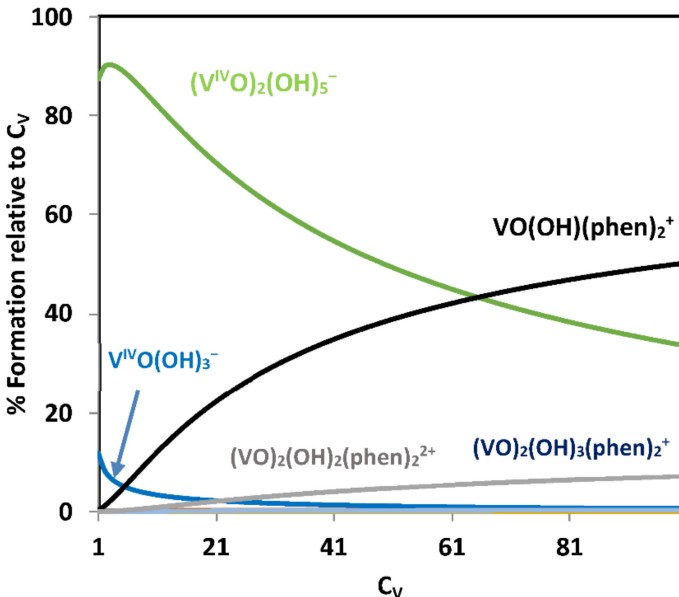

**Figure 6.** Species distribution diagrams in water at pH 7.0 for the system $V^{IV}O$-phen [97] in the concentration range $C_V$ = 1 to 100 μM ($C_V$ is the total $V^{IV}O$ concentration), with the molar ratio $V^{IV}O$:phen of 1:2. The concentrations of the dinuclear species $[(V^{IV}O)_2(phen)_2(OH)_2]^{2+}$ and $[(V^{IV}O)_2(phen)_2(OH)_3]^+$ almost coincide. The diagrams were calculated using the HySS computer program [33] including the hydrolytic species $[V^{IV}O(OH)]^+$, $[(V^{IV}O)_2(OH)_2]^{2+}$, $[(V^{IV}O)_2(OH)_5]_n^-$ and $[V^{IV}O(OH)_3]^-$ [9,22,29].

Cell culture media used in cytotoxicity studies with mammalian cells contain many potential ligands for $V^{IV}O^{2+}$, namely relatively high amounts of bovine serum albumin (BSA), mainly due to the addition of fetal bovine serum (FBS) [104,105]. In the typical case of addition of 10% FBS, the concentration of BSA is ca. 40 μM. Binding of vanadium salts and complexes to proteins has been studied, particularly aspects associated with their transport in blood [3,9,13,106–111]. Binding of vanadium compounds to serum proteins [73,108–112], including complexes containing polypyridyl co-ligands [96,97,107], have also been reported. It is important to understand that as soon as each of the $V^{IV}O$-Xphen complexes is added to the cell incubation media, they decompose, no longer being present as $[V^{IV}O(Xphen)_2]^{2+}$. This was demonstrated in [96], and we will now highlight the main points addressed.

For this purpose, we must clarify the definitions of formation constants to be used. The stability constants of complexes of $V^{IV}O^{2+}$ with phen and derivatives are defined in the usual way, considering the reaction in Equation (1) with L = phen, and the conditional binding constants of $V^{IV}O^{2+}$ to apoHTF, HSA, BSA or any other protein, at a set pH, may be defined according to Equations (10) and (11).

Regarding conditional binding (or association) constants of $[V^{IV}O(phen)_n]^{2+}$ complexes to a protein (e.g., BSA) at a set pH, these may be defined similarly to Equations (2) and (3), and in the case of phen (omitting charges) as:

$$K_1^{BC}: \quad \left[V^{IV}O(phen)\right] + BSA \leftrightarrows \left[V^{IV}O(phen)(BSA)\right] \tag{12}$$

$$K_2^{BC}: \quad \left[V^{IV}O(phen)_2\right] + BSA \leftrightarrows \left[V^{IV}O(phen)_2(BSA)\right] \tag{13}$$

The formation constants for [V$^{IV}$O(phen)(BSA)] and [V$^{IV}$O(phen)$_2$(BSA)] correspond to:

$$\beta_1^{BC}: \quad V^{IV}O^{2+} \; + \; phen \; + \; BSA \; \leftrightarrows \; \left[V^{IV}O(phen)(BSA)\right] \tag{14}$$

$$\beta_2^{BC}: \quad V^{IV}O^{2+} \; + \; 2phen \; + \; BSA \; \leftrightarrows \; \left[V^{IV}O(phen)_2(BSA)\right] \tag{15}$$

The constants defined in Equations (10)–(15) may be determined (or estimated) at predefined pH values; these are required for the speciation calculations at set pH values. The values of $K_1^{BC}$ and $K_2^{BC}$ may be related to $\beta_1^{BC}$ and $\beta_2^{BC}$ using Equations (1) and (12), to get Equation (14):

$$\log K_1^{BC} = \log \beta_1^{BC} - \log \beta_{110} \qquad \log K_2^{BC} = \log \beta_2^{BC} - \log \beta_{120} \tag{16}$$

$\beta_{110}$ and $\beta_{120}$ being the formation constants of [V$^{IV}$O(phen)]$^{2+}$ and [V$^{IV}$O(phen)$_2$]$^{2+}$ as defined in Equation (1).

In Figure 6, we depict speciation diagrams for the V$^{IV}$O-phen system at pH 7.0 in the range of total oxidovanadium(IV) concentrations ($C_V$) from 1 to 100 µM. It is clear from Figure 6 that at pH 7.0, for $C_{VO} < \sim 10$ µM most V$^{IV}$ is in the form of hydrolytic V$^{IV}$O-species (mainly [(V$^{IV}$O)$_2$(OH)$_5$]$^-$ and [V$^{IV}$O(OH)$_3$]$^-$), and that the most relevant V$^{IV}$O-phen complex at this pH is [V$^{IV}$O(OH)(phen)$_2$]$^+$, its relative importance increasing with $C_V$.

The binding constants of V$^{IV}$O$^{2+}$ to BSA were not determined, but as previously done [96], we will make the approximation that these are equal to those determined for HSA at pH 7.4 [47]. Assuming that [BSA] = 40 µM, addition of [V$^{IV}$O(phen)$_2$]$^{2+}$ in the concentration range 1–100 µM and the formation of V$^{IV}$O-BSA species, but at this stage not considering the binding of V$^{IV}$O-phen complexes to BSA, the calculated species distribution diagram (Figure S1), totally differs from that shown in Figure 6. It may be observed that V$^{IV}$O$^{2+}$ predominantly binds to BSA, and only for higher vanadium concentrations, when there is not enough BSA to bind the oxidovanadium(IV) ions present (the total BSA concentration is 40 µM), V$^{IV}$O-phen species start being relevant. We also highlight that in the same conditions most of phen is either bound to BSA or free in solution.

V$^{IV}$O-phen complexes bind to BSA but it was not feasible to adequately calculate the formation constant for the [V$^{IV}$O(phen)(BSA)] species by CD spectroscopy; the value of the constant, $K_1^{BC}$, for the binding of [V$^{IV}$O(phen)] to BSA could only be approximately estimated with this technique [96]. In order that the VO:phen:BSA species (1:1:1) might have a reasonable concentration on speciation diagrams the corresponding stability constant must be at least $\log\beta_1$((VO(phen)(BSA)) = $\log \beta_1^{BC} \sim 15$, which corresponds to a binding constant of [V$^{IV}$O(phen)] to BSA of ca. $K_1^{BC} = 10^9$. The speciation diagrams included in Figure 7 are obtained for conditions modelling the amount of BSA present in cell incubation media.

As the binding constants of all vanadium-BSA-containing species are only an approximation of their true values, the diagrams depicted in Figure 7 may only be used for semi-quantitative purposes. The conditions assumed are not equal to those of cell media used in the cytotoxicity determinations, but provide a reasonable model and allow stating that upon addition of [V$^{IV}$O(phen)$_2$]$^{2+}$ to cell culture media containing 10% FBS (where [BSA] $\approx$ 40 µM), the complex loses its integrity yielding several different species. Moreover, the type of species that have relevant concentrations depends on the amount of complex added. Additionally, as V$^{IV}$ may oxidize to V$^V$ in incubation media of cells, also forming V$^V$–phen and V$^V$–hydrolytic compounds, the complexity of the speciation of V-phen species is much higher than what Figure 7 suggests.

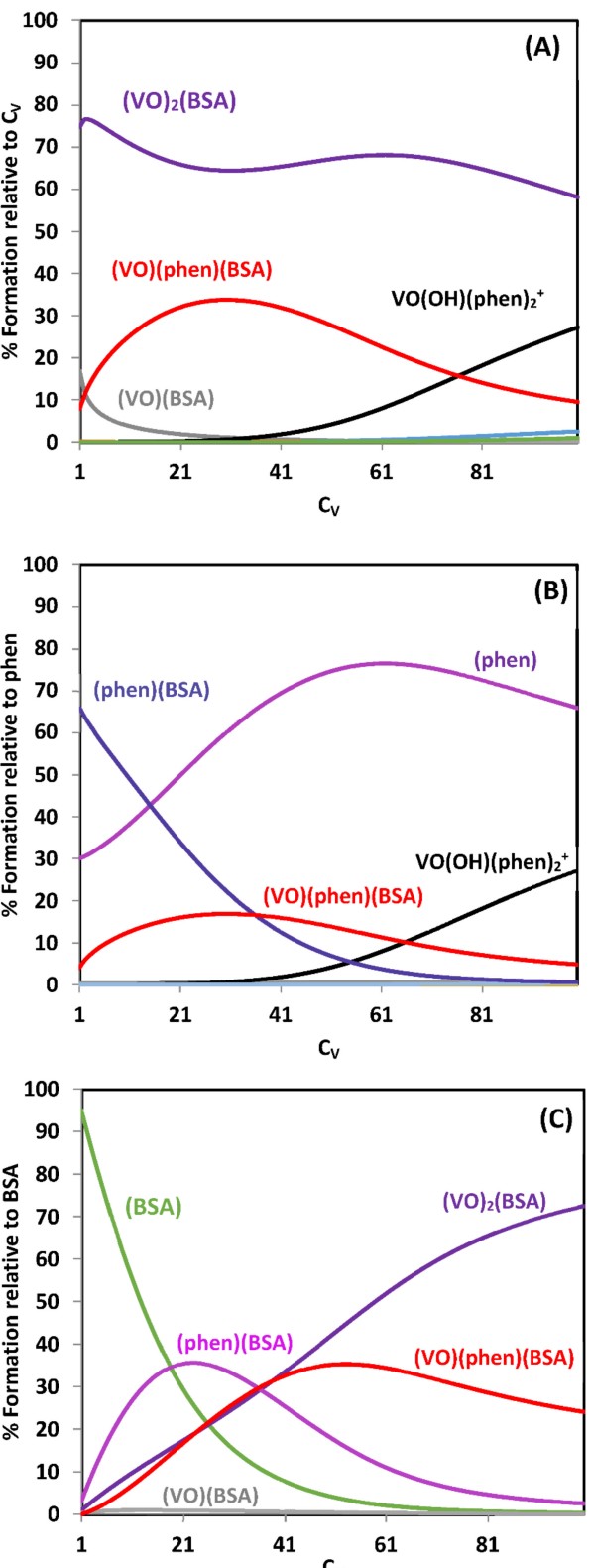

**Figure 7.** Species distribution diagrams of the system $V^{IV}O^{2+}$+phen+BSA at pH = 7 (range of $C_V$ values: 1 to 100 μM) calculated using the HySS program [33], taking a molar ratio $V^{IV}O$:phen of 1:2 and [BSA] = 40 μM. The binding of phen to BSA is taken into account with a binding constant of $5.7 \times 10^4$, and the formation of $V^{IV}O$-phen-BSA species with a stability constant of $10^{15}$ (which corresponds to $K_1^{BC} \sim 10^9$, see also text). (**A**) Representation of vanadium containing species; (**B**) Representation of 1,10-phenanthroline containing species; (**C**) Representation of BSA containing species.

Overall, it is clear that the interpretation of the obtained cytotoxicity data must take into account the speciation that the V-complexes undergo in the medium, and we cannot simply assume the existence of $[V^{IV}O(phen)_2]^{2+}$ or $[V^{IV}O(phen)]^{2+}$ either outside or inside the cells to discuss the mechanism of cytotoxic action.

Another interesting example was reported by Tshuva and co-workers [113] on a highly promising diaminotris(phenolato) $V^V$-complex that showed high hydrolytic stability in water [114] due to its six-coordination and lack of labile ligands, along with promising and broad in vitro and in vivo cytotoxic activity. The resemblance of the biological activity of the complexes and their corresponding free ligands, implying their participation as active species, inspired an investigation into the complex interactions in the cellular environment [113]. They observed: (i) the presence of free ligand inside cells, as well a $V^V$-species, which was not the initial complex; (ii) the ligand uptake seemed to be slower than the medium-formed species of the $V^V$O complex, suggesting that vanadium promoted the cell penetration and (iii) no significant impact was found for the vanadium center on the apoptotic process, which was similar for free ligand and V-complex. Overall, the studies indicated the free ligand as the active species, despite the high stability of the complexes in water, suggesting that some cellular components promote complex dissociation and that vanadium facilitates cell uptake. Like the $V^{IV}O$-Xphen system, higher activity is observed at shorter incubation periods with the complexes than with free ligands alone, highlight the role of the vanadium complex as a pro-drug.

### 2.5. Interactions of Metal Complexes with Biological Targets

Many studies reported effects of vanadium salts and complexes with biological targets such as proteins or DNA. While effects are indeed observed, misunderstandings about possible structural effects of complexes and which are the active species operating are frequent.

For example, in a study of $V^{IV}OSO_4$ and several oxidovanadium(IV) complexes with a set of six β-diketonato ligands, it was reported that $V^{IV}O$ and the complexes showed significant inhibitory activity of snake venom phosphodiesterase I enzyme. [115] All seven compounds exhibited a non-competitive type inhibition, with one complex exhibiting $IC_{50} \approx 9.8$ μM, the other six compounds (including $V^{IV}OSO_4$) exhibiting $IC_{50}$ values in the 17–33 μM range. The authors discussed the inhibitory activity mainly based on structural characteristics of the compounds, these mostly measured in organic solvents. However, the $V^{IV}O$-β-diketonato complexes certainly undergo hydrolysis and/or oxidation in aqueous solvents, as discussed above for $[V^{IV}O(acac)_2]$, and this must be taken into account in the discussion of their activity.

Another example is a study of $V^{IV}OSO_4$, $[V^{IV}O(mal)_2]$ and $[V^{IV}O(alx)_2]$ (alx = 3-hydroxy-5-methoxy-6-methyl-2-pentyl-4-pyrone), reported to act as antidiabetic agents. An in vitro Akt kinase assay was carried out using the GSK3 fusion protein, and the experiments were carried out with the complexes in 'kinase buffer' (25 mM Tris–HCl, 10 mM $MgCl_2$, 1 mM EGTA, 2 mM dithiothreitol, 0.1% BSA, pH 7.5), also containing 0.2 mM ATP and 0.5 μg of the GSK3 fusion protein substrate (EGTA = ethylene glycol bis(2-aminoethyl ether)tetraacetate, ATP = adenosine triphosphate) [116]. $[V^{IV}O(alx)_2]$ phosphorylated the Akt kinase and GSK of 3T3-L1 and GSK of adipocytes, but not $V^{IV}OSO_4$ or $[V^{IV}O(mal)_2]$. It was also reported that the vanadium uptake by the adipocytes upon 10-min stimulus with the $[V^{IV}O(alx)_2]$ and $[V^{IV}O(mal)_2]$, was $11 \pm 1.4$ and $2.5 \pm 0.2$ nmol of vanadium per $10^6$ cells, respectively [116].

These data are interesting and important, but care should be taken on elaborations of further conclusions. Besides the ligands of the complexes and the possible $V^{IV}$-oxidation, the mixture used for the Akt kinase assay contains at least two other relevant $V^{IV}O^{2+}$ binders: EGTA and BSA in quite high concentrations, therefore, a significant amount of vanadium will not be bound to allixinato or maltolato. Moreover, the vanadium uptaken and inside the adipocytes is possibly not bound to these ligands, but to other bio-ligands of the cells.

We will mention a last example. Complex *N*,*N*′-ethylenebis(pyridoxylideneiminato) vanadium(IV), [$V^{IV}O(pyren)$], originally prepared by Correia et al. [117], was reported to have relevant anticancer properties, demonstrated in in vitro tests with several human cancer cell lines such as A375 (human melanoma) and A549 (human lung carcinoma) cells. The mechanism of the effect of [$V^{IV}O(pyren)$] was assigned to apoptosis induction, triggered by ROS increase, followed by mitochondrial membrane depolarization [118].

The experiments were carried out with different concentrations of $H_2pyren$ and [$V^{IV}O(pyren)$] at 24, 48, and 72 h of incubation; pyren showed no relevant cytotoxic affect. The data obtained seems promising, but again the assignment of the active species to [$V^{IV}O(pyren)$] is elusive. It was previously reported that the ligand precursor $H_2pyren$ as well as [$V^{IV}O(pyren)$] are more soluble in water than similar salen and [$V^{IV}O(salen)$] complexes, and this allowed the determination of the formation constants in the $V^{IV}O^{2+}$ + pyren system up to pH = 5, where the neutral [$V^{IV}O(pyren)$] starts precipitating [117]. Speciations are depicted in Figure 8.

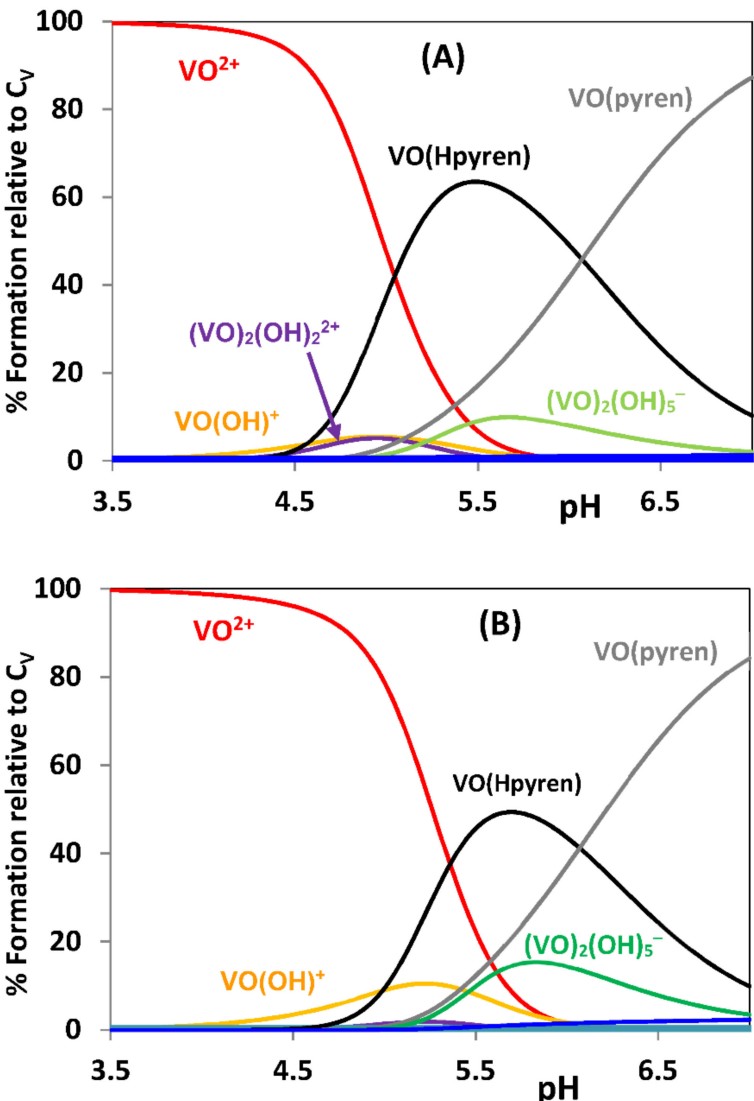

**Figure 8.** Species distribution diagrams of $V^{IV}O$-complexes formed in solutions containing $V^{IV}O^{2+}$ and $H_2pyren$, with $C_V = C_{pyren} = 100$ µM (**A**) and 10 µM (**B**), assuming the formation constants determined are valid up to pH = 7. [$V^{IV}O(Hpyren)$]$^+$ corresponds to [$V^{IV}O(pyren)$] protonated at one of the pyridine *N*-atoms. In both complexes the pyren ligand binds in a tetradentate fashion to $V^{IV}$.

The calculated concentration distribution curves of Figure 8 indicate that even at 10 μM the [$V^{IV}$O(pyren)] complex maintains its integrity. However, in solutions containing [$V^{IV}$O(pyren)] the metal progressively oxidizes and the ligand undergoes hydrolysis; after standing at 4 °C for two weeks, crystals of [$(V^{V}$O)$_2$(pyr$_{1/2}$en)$_2$] were isolated (pyr$_{1/2}$en is the half Schiff base of pyren, where one of the C=N bonds hydrolysed, see Scheme 1). This means that at lower complex concentration, both these processes may proceed much faster. Therefore, the biological properties cannot be unambiguously simply assigned to [$V^{IV}$O(pyren)], as several other $V^{IV}$ and $V^{V}$ species may be present.

**Scheme 1.** Fate of [$V^{IV}$O(pyren)] in aqueous solution and outline of some possible pathways during aging. $V_1$, $V_2$ and $V_4$ are $V^{V}$ species—see Section 2.2-and en is ethylenediamine [117].

## 3. Conclusions

It is known that the behavior of vanadium and of most first-row transition metal complexes in solution, namely in biological fluids, depend on their environment, namely on the presence of other potential ligands. In aqueous media most metal complexes undergo hydrolysis, ligand exchange and redox reactions, as well as chemical changes that depend on pH and concentration, and on the presence and nature of the several biological components present in the media. However, when reporting the biological action of metal complexes, often the possibility of chemical changes is not taken into account.

Vanadium(IV) ions are very susceptible to oxidation, and both $V^{IV}$ and $V^{V}$ ions are also very prone to hydrolysis. In this work we highlight that in the particular case of most vanadium compounds, besides oxidation and/or hydrolysis, as soon as they are dissolved in aqueous media they undergo several other types of chemical transformations and they

no longer exist in their initial form. These changes are particularly extensive at the low concentrations normally used in biological experiments. If a biological effect is observed, it is clear that in order to determine which is the active species and/or propose mechanisms of action, it is essential to evaluate the speciation of the metal-complex in the media where it is acting, but this has been neglected in most of the studies published.

Besides concentration of the metal ion, ionic strength, nature of buffer used, etc., other factors that are relevant for the hydrolytic behavior of vanadium(V) ions are time and temperature of experiments. These may influence the type and the concentration of the V-species present. Typically, chemical and spectroscopic studies are done at ca. 25 °C, while biological experiments at ~37 °C. Both these factors are relevant e.g., for the formation/decomposition of polyoxidovanadates. Moreover, we did not discuss issues associated to non-oxido vanadium(IV) complexes. Some of these complexes are quite resistant to hydrolysis, but with time they may produce oxidovanadium(IV or V) complexes, and these processes should be accounted for when discussing their biological action.

It should also be understood that in in vitro experiments, besides transformations in the incubation media, which will be relevant for the uptake process, once taken up by cells, most probably vanadium will no longer be bound to the original ligand, but to bio-ligands such as proteins of the cellular system.

If a vanadium compound is administered in vivo, either orally or by injection, the situation is much more complex. The ligand bound to the metal certainly changes during its bioprocessing, new complexes form, also with a distinct speciation profile, which may have (or not) a beneficial biological activity. The use of any metal-complex in the clinic requires the evaluation of its speciation chemistry associated with its pharmacokinetic and pharmacodynamic properties. This once more requires the evaluation of speciation of the vanadium species formed in each biological compartment and tissue involved. All these processes correspond to huge tasks and facilitate experiments and interpretations may yield misunderstandings and pseudo-conclusions that do not contribute to the advance of knowledge of the systems.

**Supplementary Materials:** The following are available online at https://www.mdpi.com/2304-6740/9/2/17/s1. Figure S1: Species distribution diagrams of the system VIVO$^{2+}$ + phen + BSA at pH = 7 (range of CV values: 1 to 100 μM) calculated using the HySS program, taking a molar ratio VIVO:phen of 1:2 and [BSA] = 40 μM. The binding of phen to BSA is taken into account with a binding constant of $5.7 \times 10^4$, but in this figure the formation of VIVO-phen-BSA species is not con-sidered.

**Author Contributions:** Conceptualization, J.C.P.; Methodology, J.C.P. and I.C.; Software (use), J.C.P. and I.C.; Validation, J.C.P. and I.C.; Formal Analysis, J.C.P. and I.C.; Investigation, J.C.P. and I.C.; Resources, J.C.P. and I.C.; Data Curation, J.C.P. and I.C.; Writing—Original Draft Preparation, J.C.P.; Writing—Review & Editing, J.C.P. and I.C.; Visualization, J.C.P. and I.C.; Supervision, J.C.P. and I.C.; Project Administration, J.C.P.; Funding Acquisition, J.C.P. and I.C. All authors have read and agreed to the published version of the manuscript.

**Funding:** This research received funding from Fundação para a Ciência e Tecnologia (FCT) projects UIDB/00100/2020), Programa Operacional Regional de Lisboa 2020. I.C. thanks program FCT Investigator (IF/00841/2012).

**Institutional Review Board Statement:** Not applicable.

**Informed Consent Statement:** Not applicable.

**Data Availability Statement:** Not applicable.

**Acknowledgments:** We thank Centro de Química Estrutural, financed by Fundação para a Ciência e Tecnologia.

**Conflicts of Interest:** The authors declare no conflict of interest.

**Abbreviations**

| | |
|---|---|
| Acac | acetylacetonate |
| alx | 3-hydroxy-5-methoxy-6-methyl-2-pentyl-4-pyrone |
| Amphen | 5-amino-1,10-phenanthroline |
| ATP | Adenosine triphosphate |
| Bipy | 2,2′-bipyridine |
| BSA | Bovine serum albumin |
| CD | Circular dichroism |
| $C_V$ | total vanadium concentration |
| dhp | 1,2-dimethyl-3-hydroxy-4(1*H*)-pyridinone |
| DMEM | Dulbecco's Modified Eagle Medium |
| DMSO | Dimethyl sulfoxide |
| EGTA | ethylene glycol bis(2-aminoethyl ether)tetraacetate |
| EPR | Electronic paramagnetic spectroscopy |
| FBS | Fetal bovine serum |
| HSA | Human serum albumin |
| $IC_{50}$ | Minimum inhibitory concentration |
| ICP-MS | Inductively coupled plasma mass spectrometry |
| Mal | maltolato |
| MEM | Minimum Essential Medium Eagle |
| $Me_2phen$ | 4,7-dimethyl-1,10-phenanthroline |
| MeOH | Methanol |
| NMR | Nuclear magnetic resonance |
| Phen | 1,10-phenanthroline |
| ROS | Reactive oxygen species |
| RPMI medium | Roswell Park Memorial Institute medium |
| SDS | Sodium dodecyl sulfate |
| Tris | Tris(hydroxymethyl)aminomethane |

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
