# Peer review of "Misinterpretations in Evaluating Interactions of Vanadium Complexes with Proteins and Other Biological Targets"

_inorganics, doi:10.3390/inorganics9020017_

Round 1
Reviewer 1 Report
In recent years, much attention has been paid to designing new vanadium complexes as a substance with promising biological properties that could in future be used in modern medicine to treat diseases of various etiology. However, it is still a significant challenge to minimize the toxic effects of vanadium compounds, which often outweigh their beneficial effects. This problem can be solved by studying the mechanism of action of selected V-complexes.
João Costa Pessoa and Isabel Correia highlight a crucial problem regarding the analysis of the relationship between the structure of the V-complexes and their physicochemical and biological properties. In this Review, they emphasize that the results of chemical studies (the vanadium complex – protein or other biological targets interactions) should be assessed very carefully as the experimental conditions leading to these results are generally very different from biological conditions. Furthermore, the active species responsible for a desire biological action are usually different that the structure of vanadium complex in its initial form.
For designing new biologically active complexes it is important to know which are the vanadium species formed in solution resulting from the dissolution of V-complex (under experimental conditions) both in vitro and in cell studies conditions. The Authors describe some important factors that govern the structure of the resulting V-complexes under experimental conditions, among which a pivotal role play a pH of a solution, concentrations of the reagents (which depends on a type of an experimental technique used), the composition of the cell incubation media as well as a buffer composition (the concertation of the buffer components is usually much more higher than the concertation of the reagents). We could also add to the above list the temperature at which the measurements are carried out (25 oC and 37oC for chemical and biological studies, respectively) as a factor which may influence the type and the concentration of the V-species.
All in all, these factors are often neglected but should be taken into account during studying structure – activity relationship.
On the other hand, we are aware that in some cases the determination of the structure of the active species is not easy due to complex redox reactions, the presence of many competitive ligands in the systems or a low solubility of the complexes.
In my opinion, the Authors presented the useful comments and valuable remarks regarding the study of the metal complex interactions with the ligands of biological importance. When the presented suggestions are taken into account during analysis experimental data it will contribute to the substantial improvement of the structure – activity relationship investigations.
I believe that the material presented in this paper will be interesting for Inorganics Readers as well as for other scientists from the field of inorganic and bioinorganic chemistry.
After the following (minor) points have been considered by the Authors, the manuscript may be accepted for publication:
- The ref. [9] is the same as [96]
- The ref. [33] is the same as [95]
- Page 6 – Eq(3): It should be mention in the text that the binding constant K2(BC) is so called conditional parameter as its value depends not only on the pH of a solution but also on the kind of the buffer solution which components (acting as competitive ligands) shown affinity to the metal ions (especially for transition ones)
- Page 7: Eq (6) – there is lack of subscripts for CP and CC
- Page 7: Give reference for Eq (6)
- Page 7: Explain I(lambda) and I(p) in Eq (7)
- I would recommend to use uM instead of M for the concentration of V-species in “species distribution diagrams”
Author Response
Comment:
For designing new biologically active complexes it is important to know which are the vanadium species formed in solution resulting from the dissolution of V-complex (under experimental conditions) both in vitro and in cell studies conditions. The Authors describe some important factors that govern the structure of the resulting V-complexes under experimental conditions, among which a pivotal role play a pH of a solution, concentrations of the reagents (which depends on a type of an experimental technique used), the composition of the cell incubation media as well as a buffer composition (the concertation of the buffer components is usually much more higher than the concertation of the reagents). We could also add to the above list the temperature at which the measurements are carried out (25 oC and 37oC for chemical and biological studies, respectively) as a factor which may influence the type and the concentration of the V-species.
Answer:
We thank the reviewer for his comments. In the section of Conclusions we added a paragraph where we also include the temperature and its influence on the type and the concentration of the V-species formed:
‘Besides concentration of the metal ion, ionic strength, nature of buffer used, etc., other factors that are relevant for the hydrolytic behavior of vanadium(V) ions are time and temperature of experiments. These may influence the type and the concentration of the V-species present. Typically, chemical and spectroscopic studies are done at ca. 25 ºC, while biological experiments at ~37 ºC. Both these factors are relevant e.g. for the formation/decomposition of polyoxidovanadates.’
Comment:
After the following (minor) points have been considered by the Authors, the manuscript may be accepted for publication:
- The ref. [9] is the same as [96] Answer: corrected
- The ref. [33] is the same as [95] Answer: corrected
- Page 6 – Eq(3): It should be mention in the text that the binding constant K2(BC) is so called conditional parameter as its value depends not only on the pH of a solution but also on the kind of the buffer solution which components (acting as competitive ligands) shown affinity to the metal ions (especially for transition ones)
Answer:
A phrase was added after eq. (3):
‘We also emphasize that the constants K1BC and K2BC correspond to the so called ‘conditional’ binding constants; their values depend not only on the pH of the solution but also on the type of buffer used. These may have components that have affinity for the metal ion, thus may act as competitive ligands.’
- Page 7: Eq (6) – there is lack of subscripts for CP and CC
Answer: This was corrected.
- Page 7: Give reference for Eq (6)
Answer: The reference was included: (P. Thordarson, Chem. Soc. Rev., 40 (2011) 1305-1323.)
- Page 7: Explain I(lambda) and I(p) in Eq (7)
Answer:
These are now defined in the text, and we include here parts of the corresponding phrases:
‘The fluorescence intensity at a given wavelength (Il)’
‘where IP and IPC are the fluorescence intensities of the biomolecule in the absence of and presence of the V-complex, respectively’
- I would recommend to use uM instead of M for the concentration of V-species in “species distribution diagrams”
Answer:
Figures were changed as suggested, using microM for the concentration of V in the XX-axis.
Reviewer 2 Report
It is a very revolutionary paper. I am sure it will have a tremendus impact on the Vanadium field. There are only minor details that will help the final version.
In-text citations are not unified, since in some cases they all appear inside the brackets separated by commas or hyphens as appropriate:
[12, 15, 71-74], and in others each quotation appears in a bracket: [3] [13] [96, 107-118].
On page 12, references 49, 63 and 57, are after the period of each paragraph.
In the case of the figures, they have a different font size, it would be good to unify it and take care of the spaces since some legends of the graph are very close to the scale.
Author Response
It is a very revolutionary paper. I am sure it will have a tremendus impact on the Vanadium field. There are only minor details that will help the final version.
Answer:
We thank the reviewer for his/her comments
We agree with the reviewer that is a revolutionary paper, but we also add that some of the main comments done are not restricted to vanadium.
In-text citations are not unified, since in some cases they all appear inside the brackets separated by commas or hyphens as appropriate:
[12, 15, 71-74], and in others each quotation appears in a bracket: [3] [13] [96, 107-118].
Answer: This was corrected.
On page 12, references 49, 63 and 57, are after the period of each paragraph.
Answer: This was corrected.
In the case of the figures, they have a different font size, it would be good to unify it and take care of the spaces since some legends of the graph are very close to the scale.
Answer:
We did our best in trying to uniformize all figures. Changes made in the XX-axis of some figures (change of M to microM values), which make these figures easier to visualize.
Reviewer 3 Report
The review reported by Joao Costa Pessoa and Isabel Correia highlights that several vanadium(IV) and vanadium(V) complexes are involved in dissociation and chemical transformation processes under the conditions of cell studies or fluorescence spectroscopy. This may lead to the misinterpretation of the pharmacokinetic or pharmacodynamic features of the complexes.
The manuscript is both well-presented and easy reading. My opinion is that the topic is interesting, falls in the scope of the journal and clearly shows, that the speciation chemistry involving the accurate determination of binding constants has a key role in the bioinorganic chemistry. Consequently, I recommend it for publication.
Minor comments:
2.1. In an attempt, a species distribution should be calculated which contains the blood serum components and their vanadium(IV) complexes. Such speciation may exhibit how the vanadium(IV) is accommodated by serum components under the conditions of biological studies.
Several non-oxido vanadium(IV) complexes are also known. Do they have any relevance in biological fluids?
2.2. Hydrolytic behavior of oxidovanadium(V) depends on the concentration of the metal ion, ionic strength etc. Hence, it should be mentioned the time factor, the formation of oligovanadates (in general the formation of polyoxoanions) is a relatively slow process.
Author Response
The manuscript is both well-presented and easy reading. My opinion is that the topic is interesting, falls in the scope of the journal and clearly shows, that the speciation chemistry involving the accurate determination of binding constants has a key role in the bioinorganic chemistry. Consequently, I recommend it for publication.
Answer: We thank the comments of the reviewer
Minor comments:
Comment 2.1:
In an attempt, a species distribution should be calculated which contains the blood serum components and their vanadium(IV) complexes. Such speciation may exhibit how the vanadium(IV) is accommodated by serum components under the conditions of biological studies.
Answer:
This is an important issue. However, we do not have enough data to proceed with this type of calculations, namely binding constants for the complexes with either albumin, transferrin, citrate, etc. Probably binding to transferrin would dominate. Moreover, several recent articles have been published by the groups of Kiss & Pessoa, Garribba and Lay addressing extensively and correctly this issue. We think that we would not add much in this particular area.
Comment:
Several non-oxido vanadium(IV) complexes are also known. Do they have any relevance in biological fluids?
Answer:
We added new phrases mentioning this in the conclusions’ section where we shortly mention non-oxido complexes:
‘Moreover, we did not discuss issues associated to non-oxido vanadium(IV) complexes. Some of these complexes are quite resistant to hydrolysis, but with time they may produce oxidovanadium(IV or V) complexes, and these processes should be accounted for when discussing their biological action.’
Comment 2.2.:
Hydrolytic behavior of oxidovanadium(V) depends on the concentration of the metal ion, ionic strength etc. Hence, it should be mentioned the time factor, the formation of oligovanadates (in general the formation of polyoxoanions) is a relatively slow process.
Answer:
We added a new paragraph mentioning this in the conclusions’ section (this is also associated a comment made by reviewer 1):
‘Besides concentration of the metal ion, ionic strength, nature of buffer used, etc., other factors that are relevant for the hydrolytic behavior of vanadium(V) ions are time and temperature of experiments. These may influence the type and the concentration of the V-species present. Typically, chemical and spectroscopic studies are done at ca. 25 ºC, while biological experiments at ~37 ºC. Both these factors are relevant e.g. for the formation/decomposition of polyoxidovanadates.’